

# Back to the Future II: Tidal evolution of four supercontinent scenarios

Hannah S. Davies[1,2], J.A. Mattias Green[3], Joao C. Duarte[1,2,4]

[1]Instituto Dom Luiz (IDL), Faculdade de Ciências, Universidade de Lisboa, Campo Grande, 1749-016, Lisboa, Portugal
[2]Departamento de Geologia, Faculdade de Ciências, Universidade de Lisboa, Campo Grande, 1749-016, Lisboa, Portugal,
[3]School of Ocean Sciences, Bangor University, Askew St, Menai Bridge LL59 5AB, UK
[4]School of Earth, Atmosphere and Environment, Monash University, Melbourne, VIC 3800, Victoria, Australia

*Correspondence to*: Hannah S. Davies (hdavies@fc.ul.pt)

**Abstract.** The Earth is currently 180 Ma into a supercontinent cycle that began with the breakup of Pangea, and will end in around 200 – 250 Ma (Mega-annum) in the future, as the next supercontinent forms. As the continents move around the planet, they change the geometry of ocean basins, and thereby modify their resonant properties. In doing so oceans move through tidal resonance, causing the global tides to be profoundly affected. Here, we use a dedicated and established global tidal model to simulate the evolution of tides during four future supercontinent scenarios. We show that the number of tidal resonances on Earth vary between 1 and 5 in a supercontinent cycle, and that they last for no longer than 20 Ma. They occur in opening basins after about 140 – 180 Ma, an age equivalent to the Present-Day Atlantic Ocean, which is near resonance for the dominating semi-diurnal tide. They also occur when an ocean basin is closing, highlighting that in its lifetime, a large ocean basin – its history described by the Wilson cycle – may go through two resonances: one when opening and one when closing. The results further support the existence of a super-tidal cycle associated with the supercontinent cycle, and gives a deep-time proxy for global tidal energetics.

## 1 Introduction

The continents have coalesced into supercontinents and then dispersed several times in Earth's history in a process known as the supercontinent cycle (Nance et al., 1988). While the cycle has an irregular period (Bradley, 2011), the breakup and reformation typically occurs over 500 – 600 Ma (Nance et al., 2013; Davies et al., 2018; Yoshida and Santosh, 2017; 2018). Pangea was the latest supercontinent to exist on Earth, forming ~300 Ma ago, and breaking up around 180 Ma ago, thus initiating the current supercontinent cycle (Scotese, 1991; Golonka, 2007). Another supercontinent should therefore form within the next 200 – 300 Ma (e.g., Scotese 2003; Yoshida, 2016; Yoshida and Santosh, 2011 and 2017; Duarte et al., 2018; Davies et al., 2018).

The supercontinent cycle is an effect of plate tectonics and mantle convection (Torsvik, 2010 and 2016; Pastor-Galan, 2018), and the breakup and accretion of supercontinents are a consequence of the opening and closing of ocean basins (Wilson, 1966; Conrad and Lithgow-Bertelloni, 2002). The life cycle of each ocean basin is known as the Wilson cycle. A supercontinent cycle may comprise more than one Wilson cycle since several oceans may open and close between the breakup and reformation of a supercontinent (e.g., Hatton, 1997; Murphy and Nance, 2003; Burke, 2011; Duarte et al., 2018; Davies et al., 2018).





As ocean basins evolve during the progression of the Wilson cycle (and associated supercontinent cycle), the energetics of the tides within the basins also change (Kagan, 1997; Green et al., 2017). Green et al. (2017; 2018) simulated the evolution of tides from the breakup of Pangea until the formation of a future supercontinent, thus spanning a whole supercontinent cycle, and found a link between Wilson cycles and tides. They also found that the unusually large present-day tides in the

Atlantic, generated because of the near-resonant state of the basin (Platzmann, 1975; Egbert et al., 2004; Green, 2010; Arbic and Garett, 2010), have only been present for the past 1 Ma. However, because the Atlantic is still spreading apart, it will eventually become too wide to sustain resonant tides in the near (geological) future. But when exactly will this happen, and is it possible that while the continents diverge and converge, other basins will reach the right size to become resonant?

The initial simulation of future tides, conducted by Green et al., (2018) using a scenario of the Earth's tectonic future presented by Duarte et al. (2018) strengthened the proof-of-concept for the existence of a super-tidal cycle associated with the supercontinent cycle. Their simulations were done using, at best, 50 Ma intervals between most of the time slices, which Green et al. (2018) suggested was not enough to resolve details of the future tidal maxima, principally, their duration.

In this work, we revisit the future evolution of Earth's tides by simulating the tide at 20 Ma intervals during the four different tectonic modes of supercontinent formation summarised by Davies et al. (2018): Pangea Ultima (based on Scotese, 2003), Novopangea (Nield, 2007), Aurica (Duarte et al., 2018) and Amasia (based on Mitchell et al., 2012). Pangea Ultima is a scenario governed by the closing of the Atlantic – an interior ocean – leading to the reformation of a distorted Pangea (Murphy and Nance 2003 and 2008 call this "closure through introversion"). Novopangea, in contrast, is dominated by the

closing of the Pacific Ocean – an exterior ocean – and the formation of a new supercontinent at the antipodes of Pangea (this is closure through extroversion; Murphy and Nance 2003). Aurica is a scenario in which the Atlantic and the Pacific close simultaneously and a new ocean opens across Siberia, Mongolia, and India, bisecting Asia (a combination scenario in which two oceans close, one by introversion and another by extroversion; Murphy and Nance 2005; Duarte et al., 2018). Finally, in the Amasia scenario, the continents gather at the North Pole, 90º away from Pangea (this is known as orthoversion; Mitchell

et al., 2012). Every scenario has the potential to develop different tidal resonances in different ocean basins at different stages in each ocean's evolution. We focus here on identifying the timing of the occurrence of resonant basins, and on mapping the large-scale evolution of tidal amplitudes and tidal energy dissipation rates in each of the investigated scenarios. We were particularly interested in understanding how common the resonant "super-tidal" states are, for how long they last, and their relationship with the Wilson cycle.

## 2 Methods

### 2.1 Tidal modelling

The future tide was simulated using the Oregon State University Tidal Inversion Software, OTIS, which has been extensively used to simulate global-scale tides of the past, present, and future (Egbert et al., 2004; Green, 2010; Green and Huber, 2013, Wilmes and Green, 2014; Green et al., 2017; 2018). OTIS was benchmarked against other software that

simulate global tides and it was shown to perform well (Stammer et al., 2014). It provides a solution to the linearized shallow water equations (Egbert et al., 2004):

$$\frac{\partial \boldsymbol{U}}{\partial t} + f \times \boldsymbol{U} = gh\nabla\big(\eta - \eta_{SAL} - \eta_{EQ}\big) - \mathbf{F} \tag{1}$$

$$\frac{\partial \eta}{\partial t} - \nabla \cdot \boldsymbol{U} = 0 \tag{2}$$





Here, **U** is the tidal volume transport vector defined as **u**$h$, where **u** is the horizontal velocity vector and h the water depth, $f$ is the Coriolis parameter, $g$ the acceleration due to gravity, $\eta$ the sea surface elevation, $\eta_{SAL}$ the self-attraction and loading elevation, $\eta_{EQ}$ the elevation of the equilibrium tide, and **F** the energy dissipation term. The latter is defined as **F** = **F**b + **F**w, where **F**b = Cd**u**|**u**| parameterises energy due to bed friction using a drag coefficient, Cd=0.003, and **F**w =C**U** represents losses due to tidal conversion. The conversion coefficient, C, is based on Zaron and Egbert (2006) and modified by Green
and Huber (2013), computed from:

$$C(x,y) = \gamma \frac{N_H \overline{N}}{8\pi\omega}(\nabla H)^2 \tag{3}$$

in Eq. (3) $\gamma$ = 50 is a dimensionless scaling factor accounting for unresolved bathymetric roughness, $N_H$ is the buoyancy
frequency ($N$) at seabed, $\overline{N}$ is the vertically averaged buoyancy frequency, and $\omega$ is the frequency of the $M_2$ tidal constituent, the only constituent analysed here. The buoyancy frequency, $N$, is based on a statistical fit to present day climatology (Zaron and Egbert, 2006), and given by N(x,y) = 0.00524exp(-z/1300), where z is the vertical coordinate counted positive upwards from the sea floor.

Each run simulated 14 days, of which 5 days were used for harmonic analysis of the tide. The model output consists of amplitudes and phases of the sea surface elevations and transports, which was used to compute tidal dissipation rates, $D$, as the difference between the time average of the work done by the tide generating force ($W$), and the divergence of the horizontal energy flux (**P**; see Egbert and Ray, 2001, for details):

$$D = W - \nabla \cdot \mathbf{P} \tag{4}$$

where W and P are given by:

$$W = g\rho\mathbf{U} \cdot \nabla(\eta EQ + \eta SAL) \tag{5}$$
$$\mathbf{P} = g\rho\mathbf{U}\eta \tag{6}$$

The orbital configuration of the Earth-Moon system was not changed during the future simulation.

**2.2 Mapping of future tectonic scenarios**

We coupled the kinematic tectonic maps produced by Davies et al. (2018) with OTIS at incremental steps of 20 Ma by using the tectonic maps as boundary conditions in the tidal model. The maps were produced using GPlates, a software specifically designed for the visualisation and manipulation of tectonic plates and continents (e.g., Qin et al., 2012; Muller et al., 2018). We used GPlates to digitise and animate a high resolution representation of present-day continental shelves and coastline (with no ice cover), created from the NOAA ETOPO1 global relief model of the Earth (see https://data.nodc.noaa.gov/cgi-
bin/iso?id=gov.noaa.ngdc.mgg.dem:316# for details). For a matter of simplification, shelf extents are kept for the full duration of the scenarios. The continental polygons do not deform, though some overlap is allowed between their margins, to simulate rudimentary continental collision and shortening. Intracontinental breakup and rifting were introduced in three of the scenarios, allowing new ocean basins to form. No continental shelves were extrapolated along the coastlines of these newly formed basin's margins. For more details on the construction of the tectonic scenarios and the respective maps, see
Davies et al. (2018).



The resulting maps were then given an artificial land mask 2° wide on both poles to allow for numerical convergence (simulations with equilibrium tides near the poles in Green et al., 2017; 2018 did not change the results there). They were then assigned a simplified bathymetry; continental shelves were set a depth of 150 m, mid-ocean ridges were assigned a depth of 1600 m at the crest point and deepening to the abyssal plains within a width of 5°. Subduction trenches were made 5800 m deep, with the abyssal plains being set to a depth maintaining present-day ocean volume. The resulting maps were averaged to a horizontal resolution of ¼° in both latitude and longitude.

We also produced and used two present-day bathymetries to test the accuracy of our results. The first – a present-day control - is based on v13 of the Smith and Sandwell bathymetry (Smith and Sandwell, 1997; https://topex.ucsd.edu/marine_topo/). A second map was then produced – the present-day degenerate bathymetry – which included a bathymetry created by using the depth values and the method described for the future slices (see Fig. 1 and corresponding description in section 2.2).

## 3 Results

The results for the present-day control simulation (Fig. 1c), when compared to the TPX09 satellite altimetry constrained tidal solution (Egbert and Erofeeva, 2002; http://volkov.oce.orst.edu/tides/tpxo9atlas.html), produced an RMS error of 12 cm. Comparing the present-day degenerate simulation (Fig. 1d) results to TPXO9, resulted in an RMS error of 13 cm. This is consistent with previous work (Green et al., 2017; 2018) and gives us a quantifiable error of the model's performance when there is a lack of topographic detail (e.g., as in our future simulations).

The present-day control simulation (Fig. 1e) has a dissipation rate of 3.3 TW, with 0.6 TW dissipating in the deep ocean. This corresponds to 137% of the observed (real) global dissipation rate (2.4 TW for M2, see Egbert et al., 2004), and 92% of the measured deep ocean rates (0.7 TW). The present-day degenerate bathymetry underestimates the globally integrated dissipation by a factor of 0.9, and the deep ocean rates by a factor of ~0.8 (Fig. 1f). The future tidal dissipation results (Fig. 3) were normalised against the degenerate present-day value (2.2 TW), to account for the bias due to underrepresented bathymetry caused when using the simplified bathymetry in the future simulations (see Green et al., 2017, for a discussion).

The resulting tidal amplitudes and associated integrated dissipation rates are shown in Figs 2, 4-6 (amplitudes), and Fig. 3 (dissipation). The latter is split into the global total rate, and rates in shallow (depths of < 500m) and deep water (depths of > 500m; Egbert and Ray, 2001), to highlight the mechanisms behind the energy loss. In the following we define a super-tide as occurring when: i) tidal amplitudes in a basin are on average meso-tidal or above, i.e., larger than 2 m, and ii) the globally integrated dissipation is equivalent to or larger than present-day values.

### 3.1 Pangea Ultima

In the Pangea Ultima scenario, the Atlantic Ocean continues to open for another 100 Ma, after which it starts closing, leading to the formation of a slightly distorted new Pangea in 250 Ma (Fig. 2 and Video S1 in the Supplementary Material). The continued opening of the Atlantic in the first 60 Ma moves the basin out of resonance, causing the $M_2$ tidal amplitude and dissipation to gradually decrease (Figs. 2 and 3a). During this period, the total global dissipation drops to below 30% of the PD rate (note that this is equivalent to 2.2 TW because we compare to the degenerate bathymetry simulation), after which, at 80 Ma, it increases rapidly to 120% of PD (Figs. 3a and 2b). This peak at 80 Ma is due to a resonance in the Pacific Ocean initiated by the shrinking width of the basin (Fig. 2b). The dissipation then drops again until it recovers and peaks around 120 Ma at 130% above PD (Fig. 3a). This second peak is caused by another resonance in the Pacific, combined with a local



resonance in the Northwest Atlantic (Fig. 2c). This period also marks the initiation of closure of the Trans-Antarctic ocean, a short-lived ocean which began opening at 40 Ma and was microtidal for its entire tenure (Fig. 2a-c). A third peak then occurs at 160 Ma, the most energetic period of the simulation, with the tides being 215% more energetic than at present due to both the Atlantic and the Pacific being resonant for $M_2$ frequencies (Fig. 2d). After this large-scale double resonance, the first

described in detail in deep-time simulations, and the most energetic relative dissipation rate encountered, the tidal energy drops, with a small recovery occurring at 220 Ma due to a further minor Pacific resonance (Fig. 2e). When Pangea Ultima forms at 250 Ma (see Fig. 2f), the global energy dissipation has decreased to 25% of the PD value, or 0.5TW (Fig. 2f).

### 3.2 Novopangea

In the Novopangea scenario, the Atlantic Ocean continues to open for the remainder of the supercontinent cycle. Consequently, the Pacific closes, leading to the formation of a new supercontinent at the antipodes of Pangea in 200 Ma (Fig. 4 and Video S2 in the Supplementary Material). As a result, within the next 20 Ma the global $M_2$ dissipation rates decrease to half of present-day values (see Fig. 3b and 4 for the following discussion). The energy then recovers to PD levels at 40 Ma as a result of the Pacific Ocean becoming resonant. From 40 Ma to 100 Ma, the dissipation rates drop, reaching

15% of the PD value at 100 Ma. There is a subsequent recovery to values close to 50% of PD, with a tidal maximum at 160 Ma due to local resonance in the newly formed East-African Ocean (Davies et al., 2018). Even though the tidal amplitude in this new ocean reaches meso-tidal levels (i.e., 2-4 m tidal range, Fig. 3b), the increased dissipation in this ocean only increases the global total tidal dissipation to 50% PD (Fig. 4e). Therefore, this ocean cannot be considered super-tidal. The tides then remain at values close to half of present-day, i.e., equal to the long-term mean over the past 250 Ma in Green et al.

(2017), until the formation of Novopangea at 200 Ma. After 100 Ma there is a regime shift in the location of the dissipation rates, with a larger fraction than before dissipated in the deep ocean (Fig. 3b).

### 3.3 Aurica

Aurica is characterized by the simultaneous closing of both the Atlantic and the Pacific Oceans, and the emergence of the new Pan-Asian Ocean. This allows allow Aurica to form via combination in 250 Ma (Fig. 5 and Video S3 in the

Supplementary Material). In this scenario, the tides remain close to present-day values for the next 20 Ma (see Fig. 3c and 5 for the results), after which they drop to 60% of PD at 40 Ma, only to rise to 114% of PD values at 60 Ma and then to 140% of PD rates at 80 Ma. This period hosts a relatively long super-tidal period, lasting at least 40 Ma as the Pacific and Atlantic go in and out of resonances at 60 Ma and 80 Ma, respectively. The dissipation then drops to 40-50% of PD, with a local peak of 70% of the PD value at 180 Ma due to resonance in the Pan-Asian Ocean. This is the same age as the North Atlantic

today, which strongly suggests that oceans go through resonance around this age. By the time Aurica forms, at 250 Ma, the dissipation is 15% of PD, the lowest of all simulations presented here.

### 3.4 Amasia

In the Amasia scenario, all the continents except Antarctica move north, closing the Arctic Ocean and forming a supercontinent around the North Pole in 200 Ma (Fig. 6 and Video S4 in the Supplementary Material). The results show that

the $M_2$ tidal dissipation drops to 60 % of PD rates within the next 20 Ma (Fig. 3d and 6 for continued discussion). This minimum is followed by a consistent increase, reaching 80% of PD rates at 100 Ma, and then, after another minimum of 40% of PD rates at 120 Ma, tidal dissipation increases until it reaches a maximum of 85% at 160 Ma. These two maxima are a consequence of several local resonances in the North Atlantic, North Pacific, and along the coast of South America, and the minimum at 120 Ma is a result of the loss of the dissipative Atlantic shelf areas due to continental collision. A major



difference between Amasia and the scenarios previously described, is that here we never encounter a full basin-scale resonance. This is because the circumpolar equatorial ocean that forms is too large to host tidal resonances, and the closing Arctic Ocean is too small to ever become resonant. However, the scenario is still rather energetic, with dissipation rates averaging around 70% of PD rates because of several local areas of high tidal amplitudes and corresponding high shelf dissipation rates (Fig. 3d).

**4 Discussion**

We investigated how the tides may evolve during four probable scenarios of the formation of Earth's future supercontinent. The results show large variations in tidal energetics between the scenarios (see Table 1 and Fig. 3), with the number of tidal maxima ranging from 1 (i.e., at present during the Amasia scenario) to 5 (including today's in the Pangea Ultima scenario) – see Table 1 for a summary. These maxima occur because of tidal resonances in the ocean basins as they open and close.

Furthermore, we have shown that an ocean basin becomes resonant for the $M_2$ tide when it is around 140 – 180 Ma old (as is the PD Atlantic). The reason for this is simple: assuming the net divergence rate of two continents bordering each side of an ocean basin is ~3 cm yr$^{-1}$ (which is close to the average drift rates today), after 140 Ma it will be 4500 km wide. Tidal resonance occurs when the basin width is half of the tidal wavelength, $L = c_gT$ (Arbic and Garrett, 2010), where $c_g=(gh)^{1/2}$ is the wave speed and T is the tidal period (here equal to 12.42 hours). For a 4000 m deep ocean, resonance thus occurs when

the ocean is around 4500 Km wide, i.e. at the age given above. This is a key result of this investigation, and again highlights the relationship between the tidal and tectonic evolution of an ocean basin. It also reiterates that ocean basins must open for at least 140 Ma to be resonant during their opening (or drift much faster that at present), e.g. the Pan-Asian ocean. If an ocean opens for less than 140 Ma, e.g. the Trans-Antarctic (80 Ma of opening) or Arctic ocean (60 Ma of opening; Miller et al., 2006), or if they drift slower than 3 cm yr$^{-1}$, they will not become resonant or even mesotidal. After this 140 Ma/4500

Km age/width threshold has been reached, the ocean may then be resonant again if it closes.

Therefore, if the geometry, and mode of supercontinent formation permits (i.e., multiple Wilson cycles are involved), several oceans may go through multiple resonances – sometimes simultaneously – as they open/close, during a supercontinent cycle. For example, during the Pangea Ultima scenario, the Atlantic and Pacific are simultaneously resonant at 160 Ma (Table 1),

and as Aurica forms, the Atlantic is resonant twice (at present, and when closing at 80 Ma), the Pacific once (closing) and the Pan-Asian ocean once (after 180 Ma, when opening; Table 1 and section 3.3).

The simulations here expand on the work of Green et al., (2018), regarding the tidal evolution of Aurica. They find a more energetic future compared to the present Aurica simulations (e.g., our Fig. 3c): their average tidal dissipation is 84% of the

PD value, with a final state at 40% of PD, whereas we find dissipation at 64% of PD on average and 15% of PD at 250 Ma. This discrepancy can be explained by two factors present in the work of Green et al. (2018): a lack of temporal resolution, and a systematic northwards displacement in the configuration of the continents, meaning their tidal maxima are exaggerated. Despite these differences, the results are qualitatively similar, and we demonstrate here that under this future scenario the tides will be even less energetic than suggested in Green et al. (2018). This, along with results from tidal

modelling of the deep past (Green et al., 2019 in review, Byrne et al., 2019 in review, Green and Hadley-Pryce personal comm. 2019, and this paper) lends further support to the super-tidal cycle concept, and again shows how strong the current tidal state is.

Present Lunar recession rates also expose the anomalous current tidal state. Retropolating present Lunar recession rates into

deep time (~3.8 cm yr $^{-1}$; Dickey et al., 1994) places lunar formation at 1.5 Ga in Earth History. However, the Moon is



geologically aged at 4.5 Ga (Kleine et al., 2005). This disparity in Lunar ages confirms that tidal dissipation rates must have been – except for occasional super-tidal periods – a fraction of the present-day value for most of deep time.

The present-day tidal anomaly represents a hurdle for deep time climate models. This can be seen in the Eocene, where the dissipation rates from Green and Huber (2013) solved a decade long problem of reconstructing a model of the Eocene
climate which conforms to the paleo-climate record (Huber et al., 2003). They found the global total tidal dissipation at 55 Ma was around 1.44 TW, with 40 % of that being dissipated in the deep Pacific Ocean, a regime significantly different to the present-day, but enough to sustain abyssal mixing that could reduce meridional temperature gradients (Green and Huber 2013).

This study represents one of many tidal modelling projects with OTIS (see Green et al., 2017; 2018, and Davies et al., 2018 for discussions) which combined, produce a suite of sensitivity simulations. Despite the fact that the method of assigning bathymetry was different here, and in other papers, and the values of $\gamma$ and $C_d$ in Eqs. (1)-(2) were changed, every result shows changes differing by no more than 10-20% of the results presented here. Further sensitivity simulations can be done in the future as our modelling capacity increases, but at this stage, we argue that our results are robust and realistic.
All four scenarios presented have an average tidal dissipation lower than the present-day, and all scenarios, except Amasia, have a series of super-tidal periods analogous to present-day. The results presented here can supplement the fragmented tidal record of the deep past (Kagan and Sundermann, 1996; Green et al., 2017) and allow us to draw more detailed conclusions about the evolution of the tide over geological time, and the link between the tide, the supercontinent cycle, and the Wilson
cycle.

## Author contribution

Hannah S. Davies: Conceptualization, Formal Analysis, Investigation, Methodology, Software, Validation, Visualization, Writing – Original draft, Writing – review & editing

J.A. Mattias Green: Conceptualization, Data Curation, Funding Acquisition, Methodology, Project Administration,
Resources, Software, Supervision, Validation, Writing – review & editing

Joao C. Duarte: Conceptualization, Funding Acquisition, Project administration, Resources, Supervision, Writing – review & editing.

## Acknowledgements

H.S. Davies acknowledges funding from FCT (ref. UID/GEO/50019/2019—Instituto Dom Luiz; FCT PhD grant ref.
PD/BD/135068/2017). J.A.M. Green acknowledges funding from NERC (MATCH, NE/S009566/1), an internal travel grant from the School of Ocean Science, and a Santander travel bursary awarded through Bangor University. J.C. Duarte acknowledges an FCT Researcher contract, an exploratory project grant ref. IF/00702/2015, and the FCT project UID/GEO/50019/2019-IDL. Tidal modelling was carried out using HPCWales and the support of Ade Fewings. We would like to thank Filipe Rosas, Pedro Miranda, Wouter Schellart, and Célia Lee, for insightful discussions and for providing
support related to several aspects of the work.

The authors declare that they have no conflict of interest.




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





**Figure Captions**

**Figure 1: a) The PD bathymetry in m.**

**b) as in a but for the degenerate PD bathymetry (see text for details).**

**c-d) The simulated M2 amplitudes (in m) for the control PD (c) and degenerate PD (d) bathymetries. Note that the colour scale saturates at 2 m.**

**e-f) as in c-d but showing M2 dissipation rates in Wm$^{-2}$.**

**Figure 2: Global $M_2$ amplitudes for six representative time slices of the Pangea Ultima scenario. The colour scale saturates at 2 m. For the full set of time slices, covering every 20 Ma see Supplementary material. Also, note that the figures presented for each**
**scenario display different time slices to highlight periods with interesting tidal signals and that the centre longitude varies between panels to ensure the supercontinent forms in the middle of each figure (where possible).**

**Figure 3: Normalised (against PD degenerate) globally integrated dissipation rates for the Pangea Ultima (a), Novopangea (b), Aurica (c), and Amasia (d) scenarios. The lines refer to total (solid line), deep (dashed line), and shelf (dot-dashed line) integrated dissipation values. Each super-tidal peak is marked where it reaches its peak, Pac = Pacific, Atl = Atlantic.**

**Figure 4: As in figure 2 but for the Novopangea scenario.**

**Figure 5: As in figure 2 but for the Aurica scenario.**

**Figure 6: As in figure 2 but for the Amasia scenario.**

**Tables**

**Table 1: Summary of the number of super-tidal peaks for each scenario.**

| Supercontinent scenario | Mode of supercontinent formation | Number of super-tidal peaks, incl. PD | Resonant basin(s), including PD | Average normalised (against PD degenerate = 2.2 TW) dissipation |
|---|---|---|---|---|
| Pangea Ultima | Introversion | 5 | Atlantic, Pacific, Pacific & Atlantic, Pacific & Atlantic, Pacific | 0.877 |
| Novopangea | Extroversion | 2 | Atlantic, Pacific | 0.520 |
| Aurica | Combination | 4 | Atlantic, Pacific, Atlantic, Pan-Asian | 0.647 |
| Amasia | Orthoversion | 1 | Atlantic | 0.723 |




**Figure 1.**






**Figure 2.**



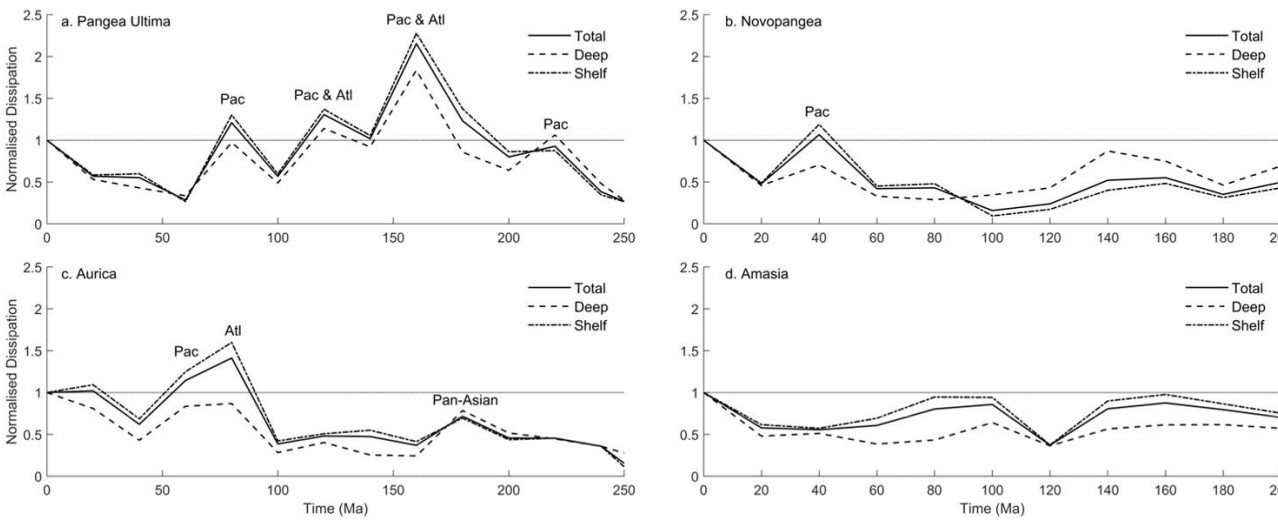


**Figure 3.**





**Figure 4.**






**Figure 5.**



Figure 6.