# Peer review of "Back to the Future II: Tidal evolution of four supercontinent scenarios"

_Earth System Dynamics, 2019_

## Referee Comment (RC1) · Daniel Pastor-Galán (Referee) · 5 Nov 2019

The paper entitled "Back to the Future II" is a well written paper which numerically models the evolution of tides into four hypothetical future scenarios where (or I should say when...) a supercontinent has formed again, namely the classical Pangea Ultima and Novo Pangea and the two more recent Amasia and Aurica models. The authors investigated the effects of the Wilson cycle on tides during the different stages of assembling such four supercontinents. Davis et al., confirmed previous studies focused in paleoclimate and also deduced from moon dynamics, tides have been much less energetic than now for most periods of time. Authors found that the moments of "supertides" (that is, similar or stronger than present day) are tightly linked with the width of oceans and therefore the Wilson cycle. Authors also found significant differences

between the 4 hypothetic supercontinents and tide evolution, which in my opinion is very significative. Considering my lack of expertise in water dynamics, I assume the code works (it has been tested before and nobody complaint), I wish other reviewers are more capable than I am and can test the modeling.

In general terms, I think it is a good research paper with some provocative ideas and deserves publication. However, after reading the paper I have questions about the work they did and I would like to see them discussed before it gets published.

1) The models that Davis et al., performed are remarkable despite the limitations and simplifications applied. However, I feel the discussion is a little limited. It verses mostly about the size (4500 Km) of ocean width. Your models seems to suggest that the shape and connections between oceans are relevant. How relevant is the shape? How relevant are the different bridges between oceans? (from a rather annular-section shape, to circular-oval and triangular?)

2) Although I understand the "catch" into modeling tides in 4 hypothetical future supercontinents, I have the feeling that a synthetic modeling in which ocean basins open and close at different angles (respect to the equator, for example) might have been more useful. In the end 3 out of 4 supercontinent models tested suggest the formation of the new supercontinent at the equator. This is, however, under debate for previous supercontinents (not only about Amasia). What would have happened in the case of opening very oblique oceans rather than equator parallel or equator perpendicular?

3) Also, the set of models do not investigate interesting settings like tethyan style oceans, mostly triangular shaped E-W oriented oceans, that have been common at least during 1.5 supercontinent cycles (Iapetus, Rheic, Paleotethys and Neotethys). What would be expected in such conditions?

4) I understand the temptation to link the supercontinent cycle with almost any long term process on Earth. At least following the 4 scenarios tested, it seems that cyclicity does not correspond with the supercontinent cycle. Results mostly imply that that the

way a supercontinent forms can change the supertidal cycles completely. And this is a great outcome!! In contrast, after reading the paper, I am convinced that tidal cycles are intrinsically linked to the Wilson cycle in 4D: when, how quick and where oceans open control the tides. Do you really think there is a supercontinent-supertide connection?

Other minor details:

Line 10: "Ma" Particularly I do not care about the debate itself on the terminology of Ma vs Myr but many (if not the majority) of geologists use Ma as "Million years ago" and prefer Myr as Mega-year to talk about time span. Also, it seems to be the recommended: -Aubry, M. P., Van Couvering, J. A., Christie-Blick, N., Landing, E., Pratt, B. R., Owen, D. E., & Ferrusquía-Villafranca, I. (2009). Terminology of geological time: Establishment of a community standard. -https://www.geosociety.org/gsatoday/archive/22/2/article/i1052-5173-22-2-28.htm#link-a

Line 29: I think Trond's and my paper suggest that it might be. Other authors are more convinced about it, but Perhaps Trond and I are among the people that think that maybe it is linked to everything and maybe it is not.

Lines 129-130: Is this error +/-12. Is it just 12 cm over or under the maximum tide? Is a +/-6? Please specify. In general I think the way the uncertainty is treated over the paper is superficial.

Lines 236-237: This is particularly interesting. Considering the particularities of supertidal periods, you should try getting a rough estimate (Fermi problem style) of how often such things had happened through Earth history... And check if that fit with our knowledge of global tectonics and moon formation etc...

Finally: "Back to the Future II" is not my favorite film from the saga, McFly.

I hope my questions and comments help the authors to improve the paper.

---

## Referee Comment (RC2) · Anonymous Referee #2 · 18 Dec 2019

General comments

The paper provides an interesting insight into the changing tidal dissipation regimes of four potential future super-continent configurations. The authors provide the results of tidal modelling at an improved temporal resolution of projected future scenarios over previous studies. The authors have found both multiple occurrences of super-tides and periods of reduced tidal dissipation (compared to the present day) which occur depending on the size of the ocean basin(s) apparent in the modeled time slices. This corroborates previous studies' conclusions that we are currently in a time of particularly energetic tidal activity, and provides insight to those who wish to examine paleoclimate. The authors also show large differences in the spatial distribution of dissipation (deep/shelf) and temporal position of super-tide events (if they occur at all) between

the different future scenarios. It is a good paper that presents an interesting result and deserves publication after some discussion. I look forward to seeing how the results presented here are built upon.

Specific comments

1) Could the authors provide some comment as to the choice of modelling only the M2 constituent (and not including K1 for example), and on why they retain the Earth-Moon orbit configuration (specifically the 12.42 hr period forcing) throughout the future simulations? Are there projections as to how this may to change within 250Ma that may be referenced? It will impact the age/size at which future ocean basin configurations form tidal resonance. This prompts a thought on the validity of the extrapolation of particular values (particularly the buoyancy frequency and ocean volume) from present day climatology for the calculation of tidal dissipation/amplitude in the future scenarios. Can the authors address these simplifications to their simulations?

2) Could the authors state clearly if equilibrium forcing is used at the pole boundaries in this study (as done in Green et al. 2018) or if vertical walls were used – the reading from Line 117 is slightly ambiguous. Since this study provides higher temporal resolution for future continent configurations from Green et al. 2018, does the equilibrium forcing (or vertical wall) at the boundary interfere with any potential tidal resonance in basins/enclosures present in this study but not present in the scenarios Green et al. 2018 considered? It is difficult to tell from the map projection used for the figures in the supplement.

3) Regarding the 4 km deep ocean calculation at Line 209: Does the average depth of any ocean basin change significantly to retain ocean volume between the four future scenarios (e.g. due to differing continent polygon overlap and/or destruction of shelves)? How applicable is this calculation of when resonance occurs for the multiple different basin shapes shown in the different scenarios?

4) The paper professes to support a link between the super-tidal and super-continent
cycles. Since each continent cycle may be comprised of one or multiple Wilson cycles (to which a super-tidal cycle seems more intrinsically linked to), is there not a lack of a well defined relationship between the period of each?

Technical corrections

line 11 – remove comma after "planet"

line 11 – Perhaps "...oceans *can* move..."

line 47 – "at best" is strange wording, perhaps "at a minimum of"

lines 75-80 – various subscripts are printed as normal sized text

line 129 – I assume "The results" refer to amplitudes. Could this be made clearer what is being compared to TPXO9.

line 152 – is PD defined in the text before its first use here?

line 246 – "...which, when combined, produce..."

I hope these comments aid the authors.

---

## Author Comment (AC1) · 22 Jan 2020

Firstly all the authors of this manuscript would like to thank Daniel Pastor-Galán for his insightful and constructive comments on the paper.

1. "I feel the discussion is a little limited. It verses mostly about the size (4500 Km) of ocean width".

The 4500 Km ocean width and the equations mentioned in the discussion represent the main conclusions of the paper. We wanted to show that we had experimentally verified the ocean width and depth required for resonance. The discussion has been expanded in the revised manuscript, because of this comment, and other comments by both reviewers.

2. Reviewer 1 mentions that a more synthetic analysis of the tide in oceans with various simplified or facsimile shapes i.e. the Tethys/annular/circular/triangular.

Synthetic modelling of different shaped oceans which close at different angles was tested, and it was deemed too much to present both the synthetic and the future supercontinent model results in one paper. Preliminary synthetic modelling was carried out by a colleague who has submitted the results in a paper to GRL. Detailed synthetic modelling (covering the ocean and continent arrangements mentioned by the reviewer in their first, second and third comments) is currently ongoing with the aim of producing sufficient results to publish in a separate paper furthering the exploration of the topic.

3. "I am convinced that tidal cycles are intrinsically linked to the Wilson cycle in 4D: when, how quick and where oceans open control the tides. Do you really think there is a supercontinent-supertide connection?"

We would still like to argue for a supercontinent-super-tide connection because even though the tide does change predominantly with the progression of the Wilson cycle, (i.e. after the ocean has opened sufficiently it will pass through at least one tidal resonance) there is still a long term trend occurring. In all scenarios we see a trend of stronger average tides during the dispersed continent phase of a supercontinent cycle, and weaker average tides during the gathered phase of the supercontinent cycle. We have clarified this idea further in the discussion section of the updated manuscript.

4. "Line 10:"Ma""

We agree that Myr is a more widespread unit and have therefore changed "Ma" to Myr" throughout the manuscript.

5. "Line 29: I think Trond's and my paper suggest that it might be. Other authors are more convinced about it, but Perhaps Trond and I are among the people that think that maybe it is linked to everything and maybe it is not."

Acknowledged, updated manuscript to include ongoing discussion

6. "Lines 129-130: Is this error +/-12. Is it just 12 cm over or under the maximum tide? Is a +/-6? Please specify. In general, I think the way the uncertainty is treated over the paper is superficial."

Root mean square error represents the standard deviation of the error, so 12 is the amount the model results deviate from the observed result of the M2 tide. This value can apply as positive or negative, either side of the "line of best fit of the data" (which in this case is measured "real world" tidal values). We have clarified the whole section presenting error and uncertainty in the updated manuscript.

7. "Lines 236-237: This is particularly interesting. Considering the particularities of supertidal periods, you should try getting a rough estimate (Fermi problem style) of how often such things had happened through Earth history... And check if that fit with our knowledge of global tectonics and moon formation etc..."

Myself and my co-authors were very intrigued by this comment and have since added a fermi/drake equation to the manuscript.

---

## Author Comment (AC2) · 22 Jan 2020

We would like to thank anonymous reviewer 2 for their helpful and constructive comments on the paper.

1. "Could the authors provide some comment as to the choice of modelling only the M2 constituent (and not including K1 for example), and on why they retain the Earth-Moon orbit configuration (specifically the 12.42 hr period forcing) throughout the future simulations? Are there projections as to how this may to change within 250Ma that may be referenced? It will impact the age/size at which future ocean basin configurations form tidal resonance. This prompts a thought on the validity of the extrapolation of particular values (particularly the buoyancy frequency and ocean volume) from present

day climatology for the calculation of tidal dissipation/amplitude in the future scenarios. Can the authors address these simplifications to their simulations?"

Modelling was carried out initially with both the M2 and the K1 constituents however the volume of data quickly became too much to present concisely in one paper. Furthermore, not only did the results for the K1 constituent corroborate the hypothesis of resonance - albeit at different wavelengths and ocean sizes - the tidal energy dissipated as a result of the K1 constituent was an order of magnitude lower than the M2 making it less impactful to the overall tidal environment.

With regards to the change in tidal, oceanographic, and orbital parameters over the period modelled. After 250 Myr, the M2 tidal period increases to 12.53 hours and lunar forcing decreases to 97%. Buoyancy frequency and ocean volume were deemed too dynamic over time to accurately predict their changes. It was therefore concluded that all parameters would be kept at present-day values. We have since added a paragraph to the results section of the manuscript presenting the change in tidal period and lunar forcing over time, and stating we kept all parameters at present-day values for all simulations.

2. "Could the authors state clearly if equilibrium forcing is used at the pole boundaries in this study (as done in Green et al. 2018) or if vertical walls were used – the reading from Line 117 is slightly ambiguous. Since this study provides higher temporal resolution for future continent configurations from Green et al. 2018, does the equilibrium forcing (or vertical wall) at the boundary interfere with any potential tidal resonance in basins/enclosures present in this study but not present in the scenarios Green et al. 2018 considered? It is difficult to tell from the map projection used for the figures in the supplement."

Equilibrium forcing at the poles was not used, vertical walls at the poles were used instead. Green (personal comm.) found that the introduction of an open boundary with an equilibrium tide as forcing does not change the results. We have clarified this part

of the methods section in the revised manuscript.

3. “Regarding the 4 km deep ocean calculation at Line 209: Does the average depth of any ocean basin change significantly to retain ocean volume between the four future scenarios (e.g. due to differing continent polygon overlap and/or destruction of shelves)? How applicable is this calculation of when resonance occurs for the multiple different basin shapes shown in the different scenarios?”

Changing ocean depth does influence the resonant width of the ocean however the deviation in abyssal ocean depth from 4 Km in the models to retain ocean volume is not large enough to change ocean resonance by a large amount. We have updated the manuscript to show that the resonant width scales with the square root of the depth so to change the resonant width of an ocean, the depth must change by a factor of four.

4. “The paper professes to support a link between the super-tidal and super-continent cycles. Since each continent cycle may be comprised of one or multiple Wilson cycles (to which a super-tidal cycle seems more intrinsically linked to), is there not a lack of a well defined relationship between the period of each?

The Supercontinent cycle, Wilson cycle, and Super-tidal cycle are secularly linked. We have clarified this link between the three cycles and their relationship with regards to the super-tidal cycle in the updated manuscript.

Technical corrections: line 11 – remove comma after “planet”

Done.

line 11 – Perhaps “...oceans *can* move...”

Clarified this sentence and mproved its structure.

line 47 – “at best” is strange wording, perhaps “at a minimum of”

Agreed, changed in manuscript.

lines 75-80 – various subscripts are printed as normal sized text

Subscripts corrected.

line 129 – I assume "The results" refer to amplitudes. Could this be made clearer what is being compared to TPXO9.

This whole section has been revised as a result of both reviewer comments

line 152 – is PD defined in the text before its first use here?

PD now defined as "Present-day" in first use.

line 246 – "...which, when combined, produce..."

Improved sentence structure.

---

## Author Response (AR1)

Author's reply – Hannah Davies – Back to the future II: Tidal evolution of four supercontinent scenarios.

Dear Editor,

Myself and my co-authors would like to thank you for your moderation of this manuscript. Through your and the reviewers' comments and guidance, we believe the manuscript is now improved from its original submission. Please find below our response to each reviewer's comments, and the changes we have made to the manuscript as a result of the input of the reviewers.

Response to reviewer 1:

Firstly, all the authors of this manuscript would like to thank Daniel Pastor-Galán for his insightful and constructive comments on the paper.

1.        "I feel the discussion is a little limited. It verses mostly about the size (4500 Km) of ocean width".

The 4500 Km ocean width and the equations mentioned in the discussion represent the main conclusions of the paper. We wanted to show that we had experimentally verified the ocean width and depth required for resonance. The discussion has been expanded as a result of this, and other comments by both reviewers.

2.        Reviewer 1 mentions that a more synthetic analysis of the tide in oceans with various simplified or facsimile shapes i.e. the Tethys/annular/circular/triangular.

Synthetic modelling of different shaped oceans which close at different angles was tested, and it was deemed too much to present both the synthetic and the future supercontinent model results in one paper. Detailed synthetic modelling (covering the ocean and continent arrangements mentioned by the reviewer in their first, second and third comments) is currently ongoing with the aim of producing enough results to publish in a separate paper.

3.        "I am convinced that tidal cycles are intrinsically linked to the Wilson cycle in 4D: when, how quick and where oceans open control the tides. Do you really think there is a supercontinent-supertide connection?"

We would still like to argue for a supercontinent-super-tide connection because even though the tide does change predominantly with the progression of the Wilson cycle, i.e. after the ocean has opened sufficiently it will pass through at least one tidal resonance, there is still a long term trend occurring. In all scenarios we see a trend of stronger average tides during the dispersed continent phase of a supercontinent cycle, and weaker average tides during the gathered phase of the supercontinent cycle. We clarified this idea further in the discussion section of the manuscript.

4.        "Line 10:"Ma""

We agree that Myr is a more widespread unit and have therefore changed "Ma" to Myr" throughout the manuscript.

5.        "Line 29: I think Trond's and my paper suggest that it might be. Other authors are more convinced about it, but Perhaps Trond and I are among the people that think that maybe it is linked to everything and maybe it is not."

Acknowledged, updated manuscript to include ongoing discussion

6.        "Lines 129-130: Is this error +/-12. Is it just 12 cm over or under the maximum tide? Isa +/-6? Please specify. In general, I think the way the uncertainty is treated over the paper is superficial."

Root mean square error represents the standard deviation of the error, so 12 is the amount the model results deviate from the observed result of the M2 tide. This value can apply as positive or negative, either side of the "line of best fit of the data" (which in this case is measured "real world" tidal values). We clarified the whole section presenting error and uncertainty in the manuscript.

7.      "Lines 236-237: This is particularly interesting. Considering the particularities of supertidal periods, you should try getting a rough estimate (Fermi problem style) of how often such things had happened through Earth history... And check if that fit with our knowledge of global tectonics and moon formation etc..."

Myself and my co-authors were very intrigued by this comment and have since added a fermi equation to the manuscript in the discussion. We have added a table 2 which summarises the total time each scenario was in a super-tidal state. We also removed two paragraphs in the discussion which were deemed less pertinent now this fermi style problem has been discussed. As a result, we believe the discussion is now more directed at discussing the results.

Response to reviewer 2:

We would like to thank anonymous reviewer 2 for their helpful and constructive comments on the paper.

1.      "Could the authors provide some comment as to the choice of modelling only the M2 constituent (and not including K1 for example), and on why they retain the Earth-Moon orbit configuration (specifically the 12.42 hr period forcing) throughout the future simulations? Are there projections as to how this may to change within 250Ma that may be referenced? It will impact the age/size at which future ocean basin configurations form tidal resonance. This prompts a thought on the validity of the extrapolation of particular values (particularly the buoyancy frequency and ocean volume) from present day climatology for the calculation of tidal dissipation/amplitude in the future scenarios. Can the authors address these simplifications to their simulations?"

Modelling was carried out initially with both the M2 and the K1 constituents however the volume of data quickly became too much to present concisely in one paper. Furthermore, not only did the results for the K1 constituent corroborate the hypothesis of resonance - albeit at different wavelengths and ocean sizes - the tidal energy dissipated as a result of the K1 constituent was an order of magnitude lower than the M2 making it less impactful to the overall tidal environment.

With regards to the change in tidal, oceanographic, and orbital parameters over the period modelled (250 Myr) we have since added a paragraph to the results section of the manuscript presenting the change in tidal period and lunar forcing over time.

2.      "Could the authors state clearly if equilibrium forcing is used at the pole boundaries in this study (as done in Green et al. 2018) or if vertical walls were used – the reading from Line 117 is slightly ambiguous. Since this study provides higher temporal resolution for future continent configurations from Green et al. 2018, does the equilibrium forcing (or vertical wall) at the boundary interfere with any potential tidal resonance in basins/enclosures present in this study but not present in the scenarios Green et al. 2018 considered? It is difficult to tell from the map projection used for the figures in the supplement."

Equilibrium forcing at the poles was not used (as done in Green et al., 2018), vertical walls at the poles were used. The introduction of an open boundary with an equilibrium tide as forcing does not change the results. Furthermore the inclusion of walls does interfere with resonance; however, it is only observed once to minimal effect, (during the Aurica scenario) nonetheless, is mentioned in the discussion. The methods mentioned in line 117 have been clarified in the revised manuscript.

3.      "Regarding the 4 km deep ocean calculation at Line 209: Does the average depth of any ocean basin change significantly to retain ocean volume between the four future scenarios (e.g. due to differing continent polygon overlap and/or destruction of shelves)? How applicable is this calculation of when resonance occurs for the multiple different basin shapes shown in the different scenarios?"

Changing ocean depth does influence the resonant width of the ocean however the deviation from 4 Km in the models to retain ocean volume is not significant to change ocean resonance by a large amount. We have updated the manuscript to show that the resonant width scales with the square root of the depth so to change the resonance of the ocean, the depth must change by a factor of four.

4.        "The paper professes to support a link between the super-tidal and super-continent cycles. Since each continent cycle may be comprised of one or multiple Wilson cycles (to which a super-tidal cycle seems more intrinsically linked to), is there not a lack of a well-defined relationship between the period of each?

The Supercontinent cycle, Wilson cycle, and Super-tidal cycle are secularly linked. We have clarified the link between the three cycles and their relationship with regards to the super-tidal cycle in the manuscript.

All technical corrections suggested by reviewer 2 have been made in the manuscript:

Technical corrections: line 11 – remove comma after "planet"

Done.

line 11 – Perhaps "...oceans *can* move..."

Clarified this sentence and improved its structure.

line 47 – "at best" is strange wording, perhaps "at a minimum of"

Agreed, changed in manuscript.

lines 75-80 – various subscripts are printed as normal sized text

Subscripts corrected.

line 129 – I assume "The results" refer to amplitudes. Could this be made clearer what is being compared to TPXO9.

This whole section has been revised as a result of both reviewer comments line 152 – is PD defined in the text before its first use here?

PD now defined as "Present-day" in first use.

line 246 – "...which, when combined, produce..."

Improved sentence structure.

[revised manuscript text omitted]

$$\text{\sout{$f_{tot}$}}N = N_{sc} \cdot N_{wc} \cdot \text{\sout{$f$}}N_{wt\text{\sout{c}}} \tag{9}$$

And:

$$\text{\sout{$T_{tot}$}}T = \frac{\text{\sout{$f_{tot}$}}N * P\text{\sout{$T$}}_{tm}}{P\text{\sout{$T$}}_{\text{\sout{$E$}}pt}} \tag{10}$$

$N_{sc}$ represents the Number of supercontinent cycles that have occurred on Earth, including the present one (we assume a minimum of 5 supercontinent cycles ; e.g. Davies et al., 2018; and references therein), $N_{wc}$ is the number of Wilson cycles per supercontinent cycle that have supported a super tide (we assume an average of 2 ), $\text{\sout{$f$}}N_{wc\text{\sout{t}}}$ is the number of tidal maxima per Wilson cycle (we assume an average of 2; this work). $P\text{\sout{$T$}}_{tm}$ is a representativetime duration of each tidal maximum (20 Myr) and $P\text{\sout{$T$}}_{Ept}$ is the age of the Earth (4.5 yr; 
[revised manuscript text omitted]

|---|---|---|---|---|
| Pangea Ultima | 5 | 730 | 100 | 13.7 |
| Novopangea | 2 | 680 | 40 | 5.9 |
| Aurica | 4 | 730 | 80 | 11.0 |
| Amasia | 1 | 680 | 20 | 2.9 |

[Figure]

**Figure 1.**

[Figure]

**Figure 2.**

[Figure]

**Figure 3.**

[Figure]

**Figure 4.**

[Figure]

**Figure 5.**

[Figure]

a. 20 Ma b. 40 Ma c. 100 Ma d. 120 Ma e. 160 Ma f. 200 Ma

Tidal Amplitude (m)

**Figure 6.**